# Influence of Effective Grain Size on Low Temperature Toughness of High-Strength Pipeline Steel

**DOI:** 10.3390/ma12223672

**Published:** 2019-11-07

**Authors:** Yanlong Niu, Shujun Jia, Qingyou Liu, Shuai Tong, Ba Li, Yi Ren, Bing Wang

**Affiliations:** 1Department of Structural Steel, Central Iron and Steel Research Institute, Beijing 100083, China1867648003@163.com (B.L.); 13867541062@163.com (B.W.); 2State Key Laboratory of Metal Materials for Marine Equipment and Applications of Iron & Steel Research Institutes of Ansteel Group Corporation, Anshan 114009, China; 138540058154@sina.com

**Keywords:** pipeline steel, toughness, cleavage unit, crack propagation, misorientation angles

## Abstract

In this study, the series temperature Charpy impact and drop-weight tear test (DWTT) were investigated, the misorientation angles among structural boundaries where the cleavage crack propagated were identified, and angles of {100} cleavage planes between adjacent grains along the cleavage crack propagated path were calculated in five directions (0°, 30°, 45°, 60°, and 90° to the rolling direction) of high-grade pipeline steel. Furthermore, the effective grain size (grain with misorientation angles greater than 15°) was redefined, and the quantitative influences of the redefined effective grain size on Charpy impact and DWTT is also discussed synthetically. The results showed that the microstructure presented a typical acicular ferrite characteristic with some polygonal ferrite and M-A islands (composed of martensite and retained austenite), and the distribution of the high-angle grain boundaries were mainly distributed in the range of 45°–65° in different directions. The Charpy impact energy and percent shear area of DWTT in the five directions increased with refinement of the redefined effective grain size, composed of grains with {100} cleavage planes less than 35° between grain boundaries. The ductile-to-brittle transition temperature also decreased with the refining of the redefined effective grain size. The redefined effective grain boundaries can strongly hinder fracture propagation through electron backscattered diffraction analysis of the cleavage crack path, and thus redefined effective grain can act as the effective microstructure unit for cleavage.

## 1. Introduction

Pipeline steels are widely used for transporting crude oil and natural gas over long distances because of their excellent combination of high strength and toughness [1,2,3,4]. At present, with the increasing global demand for energy, the exploitation of oil and gas has already extended to remote areas or oceans, which requires pipeline steel that can be used at low temperatures, as well as across more complex geological landforms. Therefore, research on the strength and toughness of pipeline steels in low temperature environments has become a hot topic [5,6]. Some investigations showed that a fully-refined acicular ferrite (AF) microstructure with some polygonal ferrite (PF) can ensure that the material has an excellent combination of strength and toughness in a low temperature environment [7,8,9].

It is known that the strength and toughness of most steels at a low temperature can be improved by grain refinement during the controlled rolling and cooling process in steel production. However, many studies have indicated that the controlled rolling and cooling technology can also result in anisotropy of properties, especially low toughness, which can restrict the optimum design of the materials [6,10,11]. Therefore, investigation into toughness anisotropy in high-strength pipeline steels is very important. However, most studies on high-strength pipeline steels are concerned with the microstructure characteristics and mechanical properties [12,13,14,15,16], while the relationship between grain size and fracture toughness in different directions is still far from being understood due to its complicated structure. The current view from some investigators is that the structure units affecting toughness are grains with grain boundary misorientation angles greater than 15° in acicular ferrite pipeline steels. In other words, the high-angle grain boundaries (HAGB) with grain boundary misorientation angles greater than 15° are the effective grains, and cracks will be arrested and then deflected by a large angle when encountering HAGB in the process of crack propagation [12,17,18]. However, a large number of experiments show that cracks can also pass straight through HAGB between grains (as shown in Figure 1). Therefore, the effective grain defined previously is limited in explaining this phenomenon in acicular ferrite steels. It is necessary to redefine the effective grain and study the relationship between the effective grain size and fracture toughness in acicular ferrite steel.

For bcc metals, the cleavage crack always propagates along {100} cleavage planes. The angles between adjacent grains will influence the ability of changing the crack propagation direction. Some investigations have studied the correlations between toughness and structure of martensitic steels by means of calculating the {100} cleavage plane angles. For example, Deng Can-Ming et al. and Shen Jun-clang et al. [19,20] found that blocks are the microstructure units controlling the cleavage fracture and ductile-to-brittle transition temperature (DBTT) of lath martensite by means of analyzing {100} cleavage plane angles. However, the relationship between toughness and structure in different directions of acicular ferrite pipeline steels by means of analyzing {100} cleavage plane angles has not yet been reported.

In this study, the complex microstructure of acicular ferrite pipeline steel was studied by means of optical microscopy (OM), scanning electron microscopy (SEM), and transmission electron microscope (TEM). The toughness in the five typical directions (0°, 30°, 45°, 60°, and 90° to the rolling direction) of acicular pipeline steel was estimated through the Charpy impact and drop weight tear test (DWTT). The misorientation angles among grain boundaries where the cleavage crack propagated were identified by electron backscattered diffraction (EBSD), and the angles of the {100} cleavage planes of adjacent grains were statistically counted. Based on the experiments above, the final purpose was to obtain the relationship between low temperature toughness and microstructure in different directions of acicular ferrite pipeline steels, and to understand the role of the boundaries during cleavage crack.

## 2. Experimental Procedure

### 2.1. Material and Chemical Composition

A commercial high-strength low alloy pipeline steel coil (used in the investigation) with a critical thickness of 21.4 mm was industrially processed on a hot strip mill equipped with an ultra-fast cooling system (for use after the rolling mill). The chemical composition of the strip is presented in Table 1.

### 2.2. Mechanical Property Tests

The tensile, Charpy, and DWTT specimens were prepared in five directions (0°, 30°, 45°, 60°, and 90° to the rolling direction), as shown in Figure 2. Tensile specimens were machined with a diameter of 5 mm and a gauge length of 25 mm. The tensile tests were performed on a CMT5105-SANS machine in accordance with SASTM A370 (Chinese standard) at room temperature with a draw speed of 1 mm/min. The Charpy specimens were in a standard V-notched geometry with a dimension of 10 × 10 × 55 mm. The Charpy tests were carried out at 20 °C, 0 °C, −20 °C, −40 °C, −60 °C, −80 °C, −100 °C, −120 °C, and −196 °C with a K-type thermocouple attached to the specimen to control the test temperature. The DWTT (305 × 75 × 21.4 mm) specimens were carried out at 20 °C, 0 °C, −15 °C, −30 °C, −45 °C, and −60 °C in accordance with the SY/T 6476 Chinese standard.

### 2.3. Microstructural Characterization and Fracture Propagation Observation

The specimens for microstructural studies were polished and etched with 4% Nital, and then the microstructures were observed using a Leica DMIRM optical microscope (OM) and a Zeiss Ultra-55 field emission scanning electron microscope (SEM, Beijing, China) equipped with an electron back-scattered diffraction (EBSD, Beijing, China) system. The specimens for EBSD were electropolished with an electrolyte containing 100 mL glacial acetic acid and 100 ml distilled water at 0.1 A for 8–10s at room temperature (about 20 °C). The scanned area for EBSD analysis was 50 × 50 μm^2^ with a step size of 0.15 μ at a magnification of 2000×, or 100 × 100 μm^2^ with a step size of 0.2 μ at a magnification of 1000×. Selected specimens were evaluated by a FEI Tecnai G2 F20 transmission electron microscope (TEM, Beijing, China) at an accelerating voltage of 200 kV. The thin foils for TEM were ground mechanically to 30–50 μm thickness and then made into 3 mm diameter discs, which were electropolished in a two-jet machine at −20 °C. Figure 3 shows the observation plane positions of the fracture propagation of Charpy impact and DWTT samples. The crack propagation behavior of Charpy impact (−196 °C) and DWTT samples (−60 °C) were also studied by OM, SEM, and EBSD after preserving the fracture surface by electroplating with nickel, and angles of the {100} cleavage plane of the grains in different directions were measured.

In order to calculate the angle of {100} cleavage planes of adjacent grains, two adjacent grains were assumed to be grain 1 and grain 2, the orientation matrix of grain 1 obtained from the EBSD test was regarded as G1, and the orientation matrix {100} cleavage plane was regarded as P. The normal vector (a_1_, b_1_, c_1_) of the {100} cleavage surface of grain 1 can be calculated by G1 × P, as demonstrated in Equation (1). Similarly, the normal vector (a_2_, b_2_, c_2_) of the {100} cleavage plane of grain 2 was deduced. The angles (φ_1_, φ_2_, φ_3_, φ_4_, φ_5_, φ_6_, φ_7_, φ_8_, φ_9_) between vectors a_1_, b_1_, c_1_, a_2_, b_2_, c_2_ was calculated by the vector formula, and the minimum angle among φ_1_, φ_2_, φ_3_, φ_4_, φ_5_, φ_6_, φ_7_, φ_8_, and φ_9_ was the {100} cleavage planes of grain 1 and grain 2.
(1)$$G_{1}=\left(\begin{array}{c} u\\ v\\ w \end{array}\begin{array}{c} r\\ s\\ t \end{array}\begin{array}{c} h\\ k\\ l \end{array}\right),P=\left(\begin{array}{c} 0\\ 0\\ 0 \end{array}\begin{array}{c} 0\\ 1\\ 0 \end{array}\begin{array}{c} 1\\ 0\\ 1 \end{array}\right),G_{1}\times P=\left(\begin{array}{c} u\\ v\\ w \end{array}\begin{array}{c} r\\ s\\ t \end{array}\begin{array}{c} h\\ k\\ l \end{array}\right)\left(\begin{array}{c} 0\\ 0\\ 0 \end{array}\begin{array}{c} 0\\ 1\\ 0 \end{array}\begin{array}{c} 1\\ 0\\ 1 \end{array}\right),\;$$

## 3. Results

### 3.1. Microstructures

Similar elongated grain structures related to the large austenite deformation rolled below the non-recrystallization temperature (Tnr) were observed, as shown in Figure 4. This can also be seen from the similar microstructure of the five directions, which mainly consisted of AF and PF with a small number of M-A Islands. AF was composed of approximately parallel slab ferrite, which combined into bundles with each other. The width of ferrite lath generally ranged from 200 to 500 nm (indicated by the arrow in Figure 5a), which would be beneficial for the strength and low temperature toughness of the material [12,21,22]. In contrast, PF is an essentially equiaxed grain structure, which concentrates at the flatted austenite grain boundary. A M-A island was formed along with the formation of acicular ferrite, which was a basic feature of acicular ferrite. The uniformly distributed M-A islands can pin grain boundaries and improve the toughness of the material [23] (indicated by the arrow in Figure 5b). The microstructure of the five directions was refined by the thermo-mechanical control process and their grain size, shown in Figure 4, ranged from 3.9 μm to 4.1μm. Figure 4 shows that the difference of microstructure morphologies, distribution, and grain size from samples with five directions were not obvious in the steel. Hence, the effect of microstructure in the different directions on anisotropic behavior can be reasonably ignored in the following study.

### 3.2. Mechanical Properties

The typical yield strength (YS) and tensile strength (TS) of high-strength low alloy pipeline steels in different directions were tested, and the results are illustrated in Table 2. The tensile tested data show that the yield strengths with different directions all exceeded 550 MPa, while their tensile strengths ranged from 640 MPa to 695 MPa, and the elongations of all the specimens were greater than 17%. The YS/TS ratios of all the specimens lie in the range of 0.83–0.87. It can also be seen from Table 2 that YS and TS are basically similar in most directions, except for the transverse direction (90° to RD). In brief, the strengths vary little in different directions.

The Charpy impact energy and percent shear area (SA%) to DWTT, varying with different directions and different temperatures, are presented in Figure 6 and Figure 7. As expected, the Charpy impact energy and SA% to DWTT all decreased with the decrease of tested temperature. The Charpy impact energy and SA% to DWTT were obviously better along the longitudinal (0° to RD) direction than other directions. The Charpy impact energy in different directions were similar, and all were above 300 J when the temperature was higher than −40 °C; and when the temperature was lower than −40 °C, the impact energy at 45° to RD and 60° to RD decreased rapidly with the decrease of temperature. Compared to the other directions, the impact toughness of the sample at 45° to RD was the worst; in other words, the decreasing trend of the impact energy of the sample at 45° to RD was the most significant one with the decrease of temperature. When the temperature was −80 °C or −100 °C, the impact energy dropped to 112 J and 25 J, respectively; while the impact energy of samples at 0°, 30°, and 90° to RD still remained 200 J or more at −80 °C, and remained 150 J or more at −100 °C. The Charpy impact energy of the sample at 60° to RD was slightly better than that at 45° to RD. Figure 6b showed that the favorable upper shelf region, the 100% ductile range in the steel, can still be attained along the direction of 30°, 60°, and 90° with the test temperature ranging from 20 °C to −60 °C. However, the favorable upper shelf region of samples at 0° to RD and 45° to RD were attained at temperatures ranging from 20 °C to −80 °C and 20 °C to −40 °C, respectively. Compared with the results in the impact energy, Figure 7 shows that the SA% to DWTT in different directions were different, except at 0 °C. Even when the temperature decreased to −15 °C, the SA% decreased significantly at 60° and 90° to RD compared to other directions. Furthermore, a brittle fracture character began to appear in the fracture morphology when the temperature dropped to −30 °C, and the SA% to DWTT at 60° to RD and 90° to RD were 48% and 42%, respectively, which illustrated that the fracture mode was brittle fracture. By contrast, the SA% to DWTT at 0°, 30°, and 45° to RD remained above 80%. When the temperature dropped to −60 °C, dimples were dominant in the fracture surface in all directions. The tendency of SA% to DWTT to change with temperature was different from that of impact energy changing with temperature because there are some complex factors that affect the DWTT, such as full-scale thickness and multi-dimensional stress [24,25].

Figure 8 shows the change in fracture mode for impact specimens in three directions (0° to RD, 45° to RD, and 90° to RD) at three temperatures (−40 °C, −60 °C, and −80 °C). It can be seen that the fractography of the samples at 45° to RD were mainly dimples at −40 °C, which indicated that the sample fractured in a ductile manner. However, the cleavage plane could be observed in the radial zone of the specimens when the temperature dropped to −60 °C and it almost covered all of the fracture surface at −80 °C. The cleavage can be clearly seen in Figure 8h, which indicates that the sample fractured in a brittle way. However, Figure 8a,c,d,f also show that the dimples of the samples at 0° to RD and 90° to RD were still dominant in the fracture surface when the test temperature was above −80 °C. This is consistent with the results obtained from Figure 6.

Figure 9 shows the change in fracture mode for DWTT specimens in three directions (0° to RD, 45° to RD, and 90° to RD) at three temperatures (0 °C, −30 °C, and −60 °C). It was expected that the ductile deformation stage would reduce, while the brittle propagation stage would extend, resulting in deterioration of the toughness with the decrease of the test temperature from 0 °C to −60 °C. The typical ductile fracture was observed at 0 °C in three directions. When the temperature dropped to −30 °C, the dominant cleavage fracture was observed at 90° to RD, large cleavage surfaces existed in it, and a small amount of ductile fracture existed. However, the ductile fracture was still the main fracture mode in 0° to RD and 45° to RD with some ductile fractures observed on the fracture surface, as shown in Figure 9d–f. The dominant cleavage fracture occurred at −60 °C in three directions.

The results discussed above show that AF and PF were obtained in all directions of the tested steels, and that the microstructure types and sizes of the test steels were similar. However, the toughness (Charpy impact energy and SA% to DWTT) of the steels and fracture behaviors in different directions were different, which indicates that the effective grain sizes of the steels affecting the low temperature toughness and fine structure in different directions were significantly different. Therefore, it was necessary to systematically research the crystal orientation relationship between the fine substructures and effective grain sizes in different directions.

### 3.3. Crystal Orientation Relation and Effective Grain Size

The distribution map of the HAGB of the high strength pipeline steel in different directions was obtained (Figure 10). It can be seen that the tendency of the frequency distribution curves of the HAGB were similar to each other and the frequencies were slightly different in different directions. The HAGB in all directions were mainly distributed in the range 45°–65°, and the frequency distribution of 55° was the strongest. More than 65% of the HAGB ranged from 45° to 65° in the five directions. This is in agreement with recent investigations on acicular pipeline steels, and the distribution of HAGB showed no significant difference in the five direction samples [26].

The acicular ferrite had a body-centered cubic (bcc) structure. Therefore, it was most likely to slip on the {110} plane [12,26,27] and the {100} cleavage angles between adjacent grains will influence the ability to change the crack propagation direction, thereby affecting the fracture toughness of the materials. Therefore, it is very important to analyze the influence of the {100} cleavage angles on the low temperature toughness of high strength pipeline steel.

Figure 11 and Table 3 are diagrams of the fracture propagation and statistics, respectively, of the crystal orientation relationship between Charpy impact and DWTT samples, which show the distribution of misorientation angles in the samples. The thin black lines marked in Figure 11 and Table 3 represent HAGB, the thick black lines represent cleavage unit boundaries, and the thin red lines represent low angle grain boundaries (LAGB) with misorientation angles lower than 15°. It can also clearly be seen that there are a number of unordered grains distributed randomly in the metallographic structure, most of which are HAGB. In addition, there are substructures in most grains, and most of the substructures are LAGB.

The orientation relationship of grains and angles between the {100} cleavage plane is obtained by measuring more than 350 fracture crack EBSD images. The average grain sizes of flattened austenite and grain size were measured to be about 200 grains. The average cleavage unit size was calibrated by measuring about 300 grains for each direction. The quantitative result of the subunit size of the microstructure in different directions is shown in Table 4. It can be seen that the flatted austenite grain size (about 16 μm) and grain size (about 4 μm) are basically the same, produced in the same process. However, the impact energy and SA% to DWTT in different directions are clearly different when tested at a low temperature. Therefore, the size of flattened austenite and grain size are not the effective size, which determines the low temperature toughness of materials.

### 3.4. Quantitative Analysis of Subunit Size and Its Influence on Performance

Table 4 and Figure 12 show that the Charpy impact at −60 °C and −80 °C, and the SA% to DWTT at −15 °C and −30 °C in different directions, all generally increase with the refining of the cleavage unit size (in the same cleavage unit, the angles of {100} cleavage planes of adjacent grains are less than 35°). In detail, the Charpy impact increased from 234 J to 343 J at −60 °C, and from 112 J to 327 J at −80 °C, and the SA% to DWTT increased from 75% to 100% at −15 °C, and 42% to 100% at −30 °C. While the cleavage unit size of Charpy impact and DWTT were refined from 16.6 μm to 13.9 μm, and 16.2 μm to 15.2 μm, respectively, indicating that the refinement of cleavage unit size makes a large contribution to the toughness of the acicular ferritic pipeline steel. Figure 13 shows the relationship between the 50% FATT (fracture appearance transition temperature) and the cleavage unit size. The 50% FATT to Charpy impact and DWTT were all strongly dependent on the cleavage unit size, with a linear relationship existing between the d^−1/2^ of the cleavage unit size and the corresponding 50% FATT to Charpy impact and DWTT. This is consistent with the Cottrell–Petch ductile–brittle fracture formula. The ductile brittle transition temperature depends on the cleavage unit size. The cleavage fracture occurs when the cleavage fracture and yield strength of the material are equal, which also fully demonstrates that cleavage unit size plays a decisive role in low temperature toughness. The cleavage unit size was thus identified as being the unit crack path for cleavage fracture. However, in previous studies, it was shown that the structural unit controlling the toughness of acicular ferrite pipeline steels is the grain size. Obviously, this point was not supported by the present study. Because Table 4 showed that the grain size in different directions was similar and had no effect on the low temperature toughness. Moreover, the size of the cleavage unit was similar to that of flattened austenite, which may indicate that the size of the cleavage unit can be refined by refining the original austenite.

Table 4 and Figure 13 show that the cleavage unit size of the impact sample was refined from 16.7 μm to 13.9 μm, and the corresponding ductile to brittle transition temperature (DBTT) to Charpy impact decreased from 405 K to 361 K. The cleavage unit size of the DWTT sample was refined from 16.7 μm to 13.9 μm, and the corresponding DBTT to Charpy impact decreased from 405 K to 361 K. The refinement of the cleavage unit size had a significant influence on decreasing the DBTT of the material, indicating that the refinement of the cleavage unit contributes more to the cleavage fracture strength than to the yield strength. Therefore, the refinement of cleavage unit size resulted in a decrease in DBTT. According to Griffith theory, the critical fracture stress is expressed as Equation (2) [28].
(2)$$\delta_{f}={\left[\frac{\pi E\gamma_{P}}{(1-\nu^{2})d}\right]}^{1/2}$$
where *δ_f_* is critical fracture stress, *E* is the Young’s modulus, *γ_p_* is the plastic deformation energy, ν is Poisson’s ratio, and d is the effective grain size, which can also be regarded as the cleavage unit size in this study as the cleavage fracture unit is determined as being the cleavage unit.

From the mechanical aspect, the fracture appearance transition behavior can be interpreted as a result of the competition between the yield strength and fracture strength. The effect of temperature on them is different, which leads to a fracture appearance transition of low alloy steel with a body-centered cubic. Generally, the yield strength increases rapidly with a decrease in temperature, while the fracture stress is independent of temperature. At the temperature ranges in which the fracture stress is lower than the yield strength, as the ductility is caused by the slip motion of dislocation, the resistance of dislocation motion increases with a decrease in temperature. Therefore, the yield strength increases with a decrease in temperature, while the cleavage fracture strength is less affected by temperature. When the temperature is lower than the fracture appearance transition temperature and the yield strength is higher than the cleavage fracture strength, the fracture mode changes from a microporous aggregated ductile fracture to a cleavage brittle fracture, and the fracture surfaces will reveal cleavage because cleavage fractures occur before plastic deformation. Obviously, its low temperature toughness is poor.

The toughness of the materials is influenced by the crack initiation and crack propagation path. Fracture theory suggests that the initiation of a crack mainly relies on the degree of stress concentration. In the process of crack propagation of the Charpy impact and DWTT samples, dislocation piles up at the secondary phases, such as M-A islands, coarse cementite, or nonmetallic inclusions, inducing the stress to concentrate at second phases, or the interface between second phases and matrix. As non-metallic inclusions are well controlled in the pipeline strip, cracks do not deflect sharply when they encounter non-metallic inclusions generally. The critical fracture stress is related to the fracture path and frequency of crack propagation in a sharp deflection. Moreover, it is known that the smaller the cleavage unit size is, the higher the frequency of sharp crack deflection is. In addition, the crack propagation path can be effectively hindered by deflecting the crack propagation path during the Charpy impact and DWTT test as the size of the cleavage unit decreases.

### 3.5. Cleavage Crack Propagation Path

Figure 14 and Figure 15 are the typical actual crack propagation paths of Charpy impact and DWTT specimens obtained by EBSD analysis. The zone surrounded by grains with {100} cleavage planes lower than 35° represents a cleavage unit. It showed that the cleavage cracks crossed HAGB directly (Figure 14, 2–11, in Charpy samples, and Figure 15, 1–9, in DWTT samples) or only occurred as frequent small deflections at HAGB in the same cleavage unit. Only when the crack extended to the boundary of two cleavage units (Figure 14, 6–7 and Figure 15, 5–6), a great deflection occurred, and more energy was needed for the cleavage crack to cross the cleavage unit boundary. The experimental data was consistent with the result discussed in Figure 11. Obviously, grains 6 and 7 in Figure 14 and grains 5 and 6 in Figure 15 belonged to different cleavage units. The angles of {100} cleavage planes between adjacent cleavage units were all above 35°. When the cleavage crack encountered the grain boundary of 6 and 7 in Figure 14 or 5 and 6 in Figure 15, it was arrested and then deflected by a large angle, and a sharp deflection occurred along the propagation direction of the crack, which effectively retards the propagation speed of the crack. Because the positive bonding strength of the {100} crystal plane of body-centered cubic metals was lower than that of other crystal planes, when the stress exceeded the cleavage fracture strength, the cleavage fracture crack first propagated along the {100} crystal plane rapidly. When the orientation of different {100} crystal planes changed, the cleavage crack propagation was hindered. As a result, more energy was used for the cleavage crack to cross the grain boundaries with {100} cleavage plane angles of adjacent grains greater than 35°, compared to those less than 35°. The cleavage unit was thus further identified as the minimum effective structural unit for affecting the low temperature toughness of materials.

## 4. Conclusions

The toughness, including the Charpy impact, drop-weight tear test (DWTT), and crack propagation behaviors in five different directions were investigated in detail in this study, and the different crystallographic relationships and angles between the {100} planes were obtained. Scanning electron microscopy (SEM), transmission electron microscopy (TEM), and electron backscattered diffraction (EBSD) results and fracture behaviors were applied to analyze the relationships among direction, the cleavage unit, and the low toughness of Charpy impact and DWTT properties. The following conclusions are drawn from the research.The microstructure in different directions all consists of refined acicular ferrite (AF) and polygonal ferrite (PF). The distribution of high angle grain boundaries (HAGB) was similar and mainly distributed in the range of 45°–65°, which can ensure acicular ferrite has excellent toughness. However, they have no significant influence on the anisotropy of toughness in acicular pipeline steel.The impact energy and SA% to DWTT in the five directions increased when the cleavage unit was composed of grains with a {100} cleavage plane less than 35° between grain boundaries, and the ductile to brittle transition temperature decreased. The cleavage unit composed of grains with a {100} cleavage plane less than 35° between grain boundaries was identified as the effective grain for cleavage fracture.In the actual process of crack propagation, when the cleavage crack encounters another cleavage unit (refined redefined effective grain), it will be arrested and then largely change its propagation direction, while when it encounters HAGB, it is possible to cross the HAGB directly. Therefore, it is important to refine the redefined effective grain size to improve the low temperature toughness of materials.

## Figures and Tables

**Figure 1 materials-12-03672-f001:**
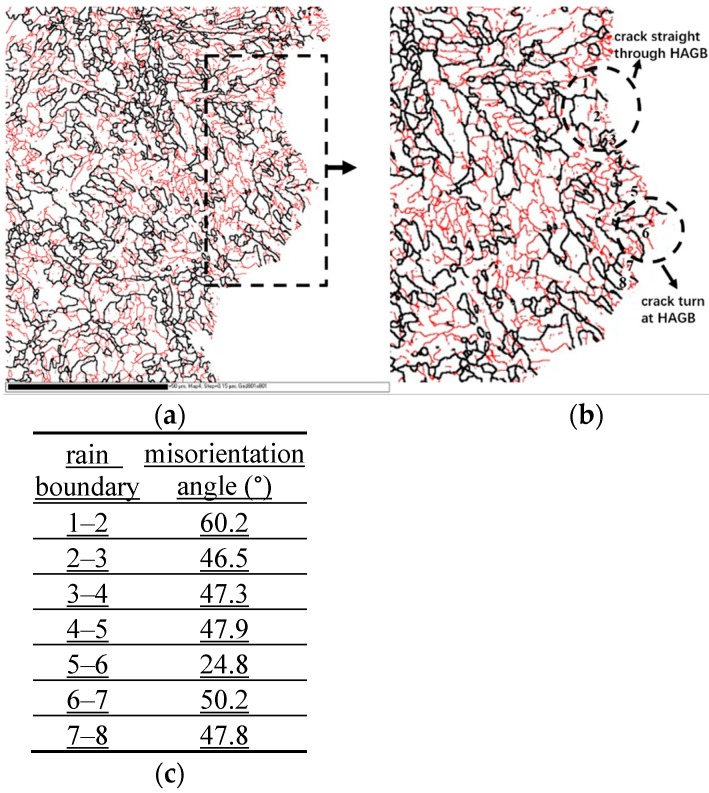
(**a**) Boundary distribution along the crack propagation path; (**b**) enlarged image of the dotted wire frame in Figure 1a; (**c**) misorientation angle between the grain boundaries.

**Figure 2 materials-12-03672-f002:**
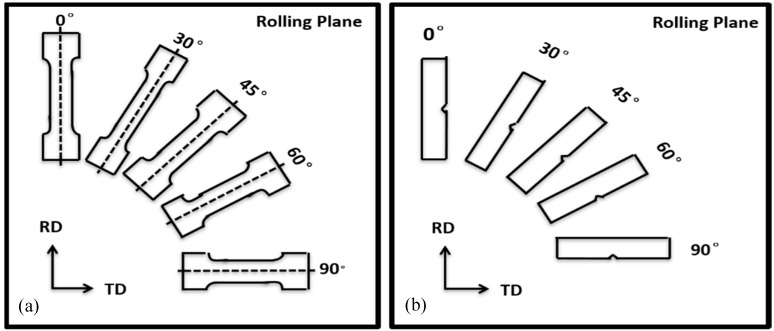
Orientation of the mechanical test specimens relative to the steel coil, where RD and TD stand for the rolling and transverse directions respectively: (**a**) directions of tensile test specimens and (**b**) different orientation of the Charpy impact and DWTT specimens.

**Figure 3 materials-12-03672-f003:**
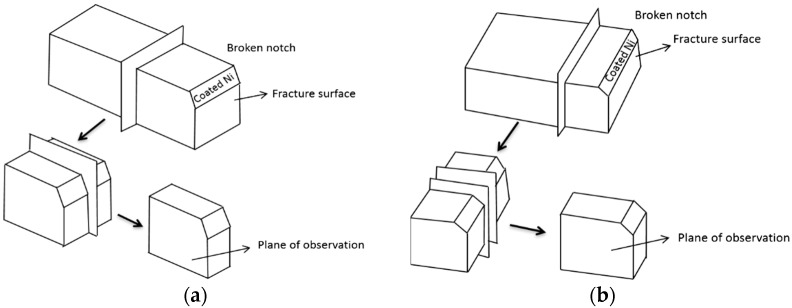
Observation plane positions of fracture propagation for Charpy impact and DWTT samples: (**a**) fracture propagation plane of Charpy impact, (**b**) fracture propagation plane of DWTT.

**Figure 4 materials-12-03672-f004:**
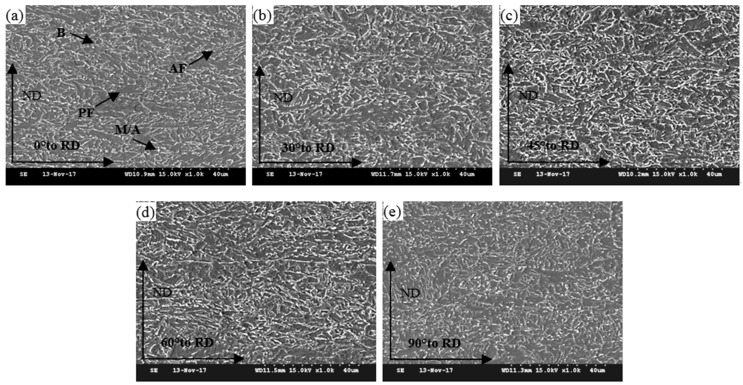
Scanning electron micrographs of samples in five directions: (**a**) 0° to RD, (**b**) 30° to RD, (**c**) 45° to RD, (**d**) 60° to RD, and (**e**) 90° to RD.

**Figure 5 materials-12-03672-f005:**
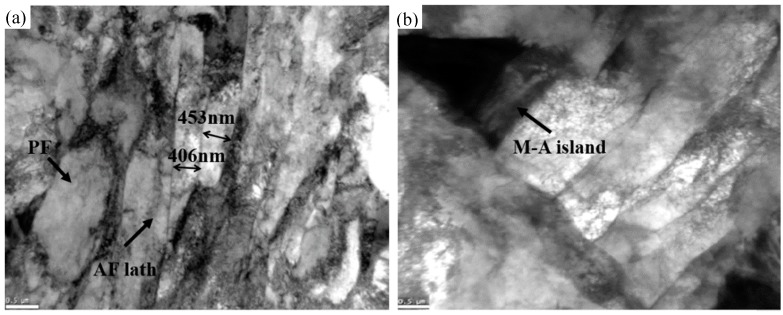
Transmission electron micrographs: (**a**) AF and PF, (**b**) M-A islands.

**Figure 6 materials-12-03672-f006:**
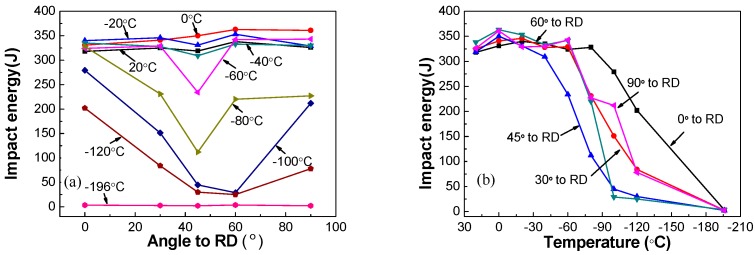
Charpy impact test result: (**a**) curve between impact energy and directions, (**b**) curve between impact energy and temperature.

**Figure 7 materials-12-03672-f007:**
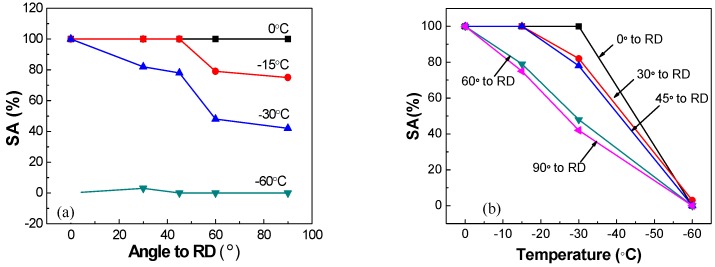
DWTT results: (**a**) curve between SA% (percent shear area) and directions, (**b**) curve between SA% and temperature.

**Figure 8 materials-12-03672-f008:**
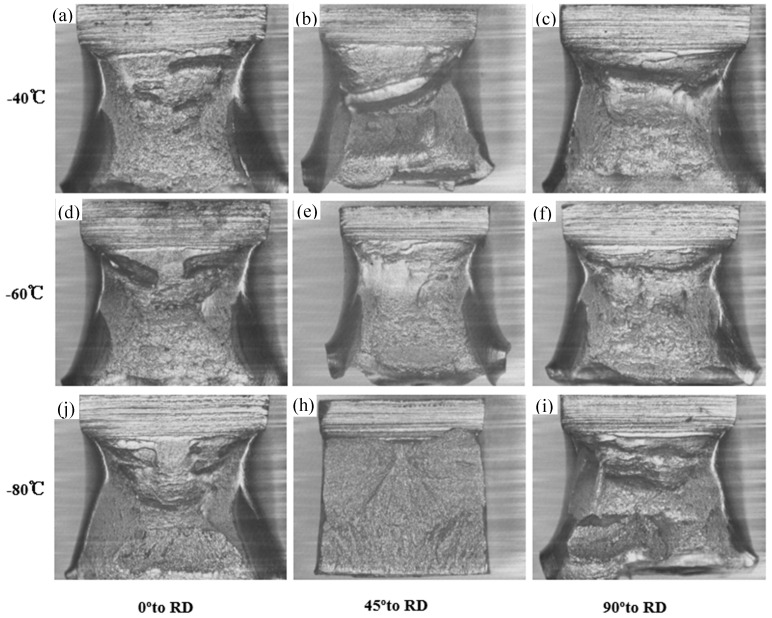
Selected fracture surfaces of Charpy specimens in three orientations (0° to RD, 45° to RD and 90° to RD) at three temperatures (−40 °C, −60 °C, and −80 °C): (**a**)−40 ℃ and 0° to RD, (**b**) −40 °C and 45° to RD, (**c**) −40°C and 90° to RD, (**d**) −60°C and 0° to RD, (**e**)−60 °C and 45° to RD, (**f**) −60°C and 90° to RD, (**j**) −80 °C and 0° to RD, (**h**) −80°C and 45° to RD, (**i**) −80 °C and 90° to RD.

**Figure 9 materials-12-03672-f009:**
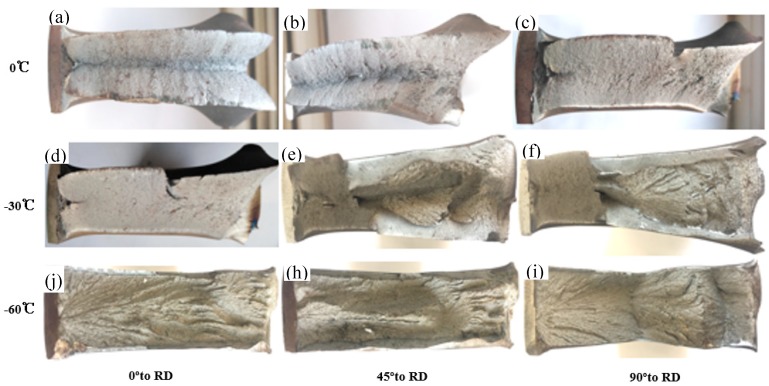
Selected fracture surfaces of the three directions (0° to RD, 45° to RD, and 90° to RD); DWTT specimens at three temperatures (0 °C, −30 °C, and −60 °C): (**a**) 0 °C and 0° to RD, (**b**) 0 °C and 45° to RD, (**c**) 0 °C and 90° to RD, (**d**) −30 °C and 0° to RD, (**e**) −30 °C and 45° to RD, (**f**) −30 °C and 90° to RD, (**j**) −60 °C and 0° to RD, (**h**) −60 °C and 45° to RD, (**i**) −60 °C and 90° to RD.

**Figure 10 materials-12-03672-f010:**
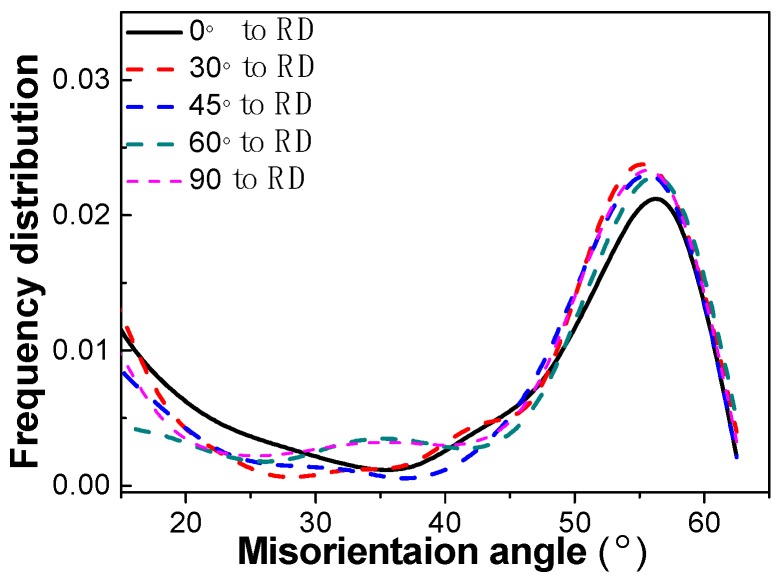
Angle boundaries distribution map of the pipeline steel in different directions.

**Figure 11 materials-12-03672-f011:**
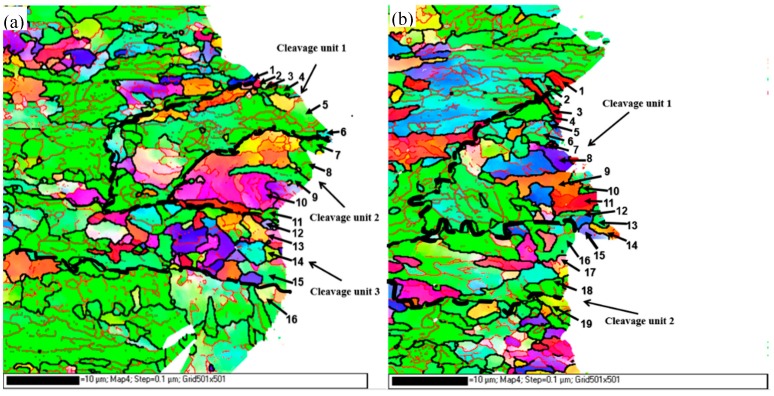
Schematic diagram of the fracture propagation of Charpy impact and DWTT samples: (**a**) Charpy impact, (**b**) DWTT.

**Figure 12 materials-12-03672-f012:**
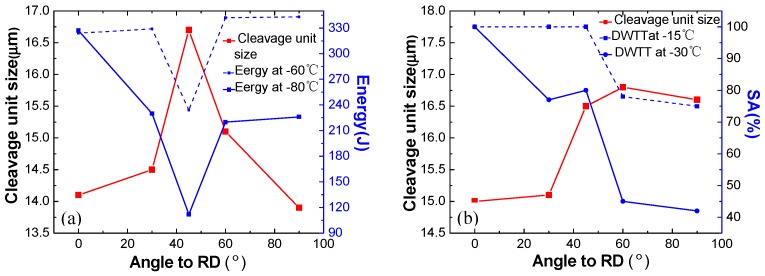
Relationship between toughness and cleavage unit size in different directions: (**a**) curve between impact energy and cleavage unit size, (**b**) curve between SA% and cleavage unit size.

**Figure 13 materials-12-03672-f013:**
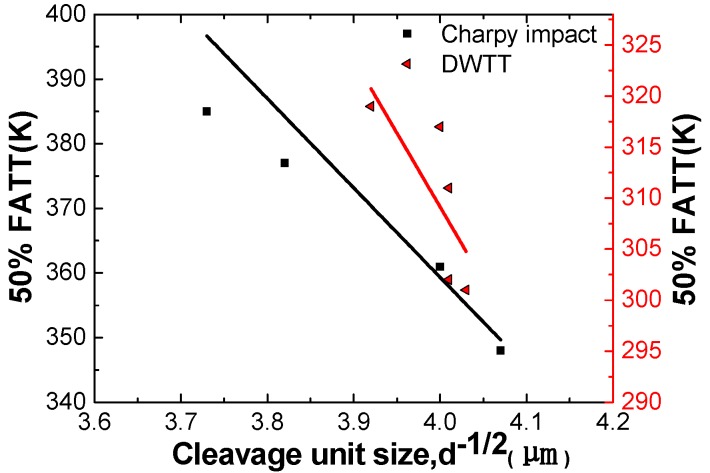
Relationship between cleavage unit size and 50% DBTT (Charpy impact and DWTT).

**Figure 14 materials-12-03672-f014:**
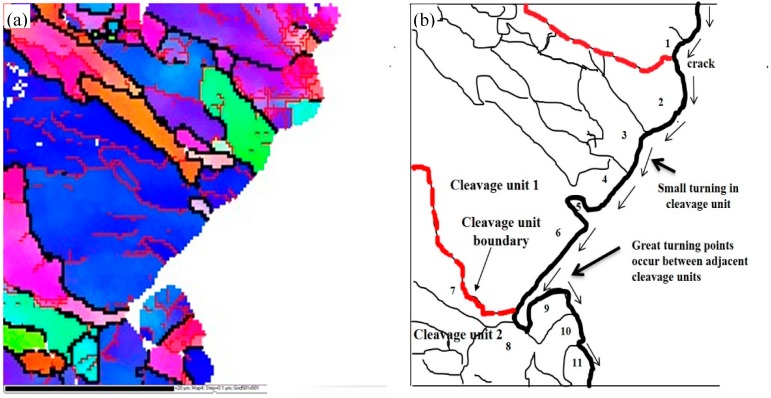
Orientation and schematic map of Charpy impact crack propagation: (**a**)orientation map of Charpy impact crack propagation, (**b**) schematic map of Charpy impact crack propagation according to Figure 14a.

**Figure 15 materials-12-03672-f015:**
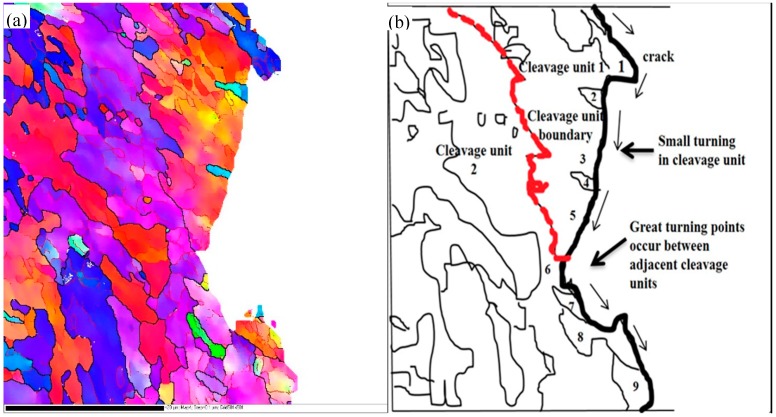
Orientation and schematic map of DWTT crack propagation: (**a**)orientation map of DWTT crack propagation, (**b**) schematic map of DWTT crack propagation according to Figure 15a.

**Table 1 materials-12-03672-t001:** Chemical composition of the studied steel (wt %).

Element	C	Si	Mn	S	P	Nb	Mo	V	Ti	Ni	Cr	Cu
Content	0.06	0.2	1.75	0.001	0.007	0.075	0.23	0.022	0.014	0.21	0.25	0.01

**Table 2 materials-12-03672-t002:** Mechanical proprieties of high-strength pipeline steel in different directions.

	Directions	YS (Rp_0.2_) MPa	TS MPa	YS/TS (%)	A%	Section Shrinkage (%)
21.4 mm	0°	560	675	0.83	27.5	81
	30°	555	640	0.87	31	80
	45°	565	655	0.86	27	85
	60°	560	670	0.84	26.5	84
	90°	600	695	0.86	23.5	80

**Table 3 materials-12-03672-t003:** Statistics of the crystal orientation relationship between Charpy impact and DWTT samples: Charpy impact corresponding to Figure 11a,b DWTT corresponding to Figure 11b.

Charpy Impact				DWTT			
Position	Grain	Misorientation Angle	{100} Planes Angle	Position	Grain	Misorientation Angle	{100} Planes Angle
cleavage unit 1and adjacent grain	1–2	48.9	40.23	cleavage unit 1 and adjacent grain	1–2	55.26	20.12
cleavage unit 1	2–3	47.5	12.25	cleavage unit 1	2–3	54.12	41.23
	3–4	46.3	14.62		3–4	58.31	20.43
	4–5	54.1	17.59		4–5	58.22	30.23
	5–6	48.7	12.34		5–6	52.33	25.32
cleavage unit2	6–7	56.3	35.62		6–7	37.22	17.24
	7–8	52.1	21.68		7–8	52.56	38
	8–9	51.6	22.17		8–9	49.37	10.34
	9–10	54.1	20.96		9–10	37.45	15.24
	10–11	52.5	42.24		10–11	25.08	20.18
cleavage unit 3	11–12	58.4	12.11		11–12	52.12	19.15
	12–13	54.1	35.24		12–13	55.15	21.55
	13–14	55.2	32.17		13–14	36.78	18.45
	14–15	54.3	32.48		14–15	60.25	40.4
cleavage unit 3 and adjacent grain	15–16	52.6	45.19	cleavage unit 2	15–16	62.12	44.87
					16–17	55.31	25.12
					17–18	54.32	37.25
				cleavage unit 2 and adjacent grain	18–19	49.22	40.21

**Table 4 materials-12-03672-t004:** Statistical diagram of microstructure unit size and toughness.

Project	0° to RD	30° to RD	45° to RD	60° to RD	90° to RD
Flatted austenite grain size (μm)	15.8	16.1	15.8	15.9	16.0
Cleavage unit size of impact sample (μm)	14.1	14.5	16.7	15.1	13.9
Cleavage unit size of DWTT sample (μm)	15	15.1	16.5	16.8	16.6
Grain size (μm)	4.0	4.1	3.9	4.2	4.1
50% FATT (K)- impact sample	405	377	348	361	385
50% FATT (K)-DWTT sample	317	319	311	301	302
−60 °C impact energy (J)	324	329	234	342	343
−80 °C Charpy energy (J)	327	230	112	220	226
−15 °C SA (%) to DWTT	100	100	100	78	75
−30 °C SA (%) to DWTT	100	77	80	45	42

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
