# Peer review of "Influence of Effective Grain Size on Low Temperature Toughness of High-Strength Pipeline Steel"

_materials, 2019, doi:10.3390/ma12223672_

Round 1
Reviewer 1 Report
The report contains interesting measurements and analysis, in the latter case as far as the description goes. Specific comments are given as follows:
There are mis-spellings, such as "angel" in the abstract and "Testes" in the Section 2.2 heading. A number of words appear to be mis-used , such as "redefined" and "synthetically". On page 11, it should be slip on {110}. The important Figure 13 should be associated with Cottrell-Petch and related analyses for the grain size dependence of the fracturing transition; this consideration also relating to updated references should be obtained past the Griffith equation (2). Figures 14 and 15 are very much appreciated as an excellent tracking of the orientations of fracturing paths at the grain size level; and in this regard, the authors might refer to the article by Pineau: "Crossing grain boundaries ... by ... cleavage .... cracks", Phil. Trans. R Soc A 373:20140131 (2014), plus others.Author Response
Please see the attachment.

Reviewer 2 Report
The authors investigated Charpy impact and DWTT properties of high strength pipeline steel at various temperatures and the microstructure of the tested specimen. They revealed that not the grain size but the cleavage unit size is dominant factor to determine low temperature toughness, which would be of interest to the readers. I would recommend it for acceptance after minor revision. My comments and questions are listed below:
Line 14, "M-A islands" appears without any explanation of the meaning of abbreviation before this line. Line 35, "Although" should be deleted. Line 52, "redefined" should be "redefine". Line 202, "Fig. 9 (h)" should be "Fig. 8 (h)". "Fig. 8 and Fig. 9" should be "Fig. 8 (a), (c), (d) and (f)". Line 206, "Fig. 7" should be "Fig. 6". Line 211, "(0 °C, -15 °C and 30 °C)" should be "(0 °C, -30 °C and 60 °C)". Sec. 3.4, the definition of cleavage unit size is not clearly shown. Which lengths were averaged? Sec. 3.4, I'm wondering why the cleavage unit of Charpy specimen is different from that of DWTT. Explaining the difference and possible reasons would be interest to the readers. Sec. 3.4, is it possible to calculate the redefined effective grain size before Charpy or DWTT testing?
